# Chemical Oral Cancerogenesis Is Impaired in PI3Kγ Knockout and Kinase-Dead Mice

**DOI:** 10.3390/cancers13164211

**Published:** 2021-08-21

**Authors:** Giovanni Nicolao Berta, Federica Di Scipio, Zhiqian Yang, Alessandra Oberto, Giuliana Abbadessa, Federica Romano, Maria Elisabetta Carere, Adriano Ceccarelli, Emilio Hirsch, Barbara Mognetti

**Affiliations:** 1Department of Clinical and Biological Science, University of Turin, Regione Gonzole 10, 10043 Orbassano, TO, Italy; federica.discipio@unito.it (F.D.S.); giuliana.abbadessa@unito.it (G.A.); mariaelisabetta.carere@unito.it (M.E.C.); adriano.ceccarelli@unito.it (A.C.); 2Scientific Research Center, First Affiliated Hospital of Guangdong Pharmaceutical University, No. 19 Nonglinxia Road, Guangzhou 510080, China; zhiqian_yang@gdpu.edu.cn; 3Department of Neuroscience, University of Turin, Regione Gonzole 10, 10043 Orbassano, TO, Italy; alessandra.oberto@unito.it; 4Neuroscience Institute of the Cavalieri-Ottolenghi Foundation, Regione Gonzole 10, 10043 Orbassano, TO, Italy; 5Department of Surgical Sciences, C.I.R. Dental School, University of Turin, 10126 Turin, Italy; federica.romano@unito.it; 6Department of Molecular Biotechnology and Health Sciences, University of Turin, Via Nizza 52, 10126 Turin, Italy; emilio.hirsch@unito.it; 7Department of Life Science and System Biology, University of Turin, Via Accademia Albertina 13, 10123 Turin, Italy

**Keywords:** PI3Kγ, chemical carcinogenesis, 4NQO, transgenic mice, oral squamous cell carcinoma

## Abstract

**Simple Summary:**

Oral carcinoma remains one of the most challenging cancers to be cured and the pharmacological approach is often ineffective. The identification of novel molecular targets will greatly improve its management. We wondered if PI3Kγ might be looked at as a target in oral cancer handling. In this preclinical study, we analyzed the role of PI3Kγ in a murine transgenic model. We demonstrated that the absence/inhibition of PI3Kγ might be a reasonable strategy to impair the development of preneoplastic and neoplastic lesions of the oral cavity. PI3Kγ is not required for normal development, life span, or basic immune responses, unless under stress conditions; therefore, low toxicity and few side effects are expected by acting on PI3Kγ as a therapeutic target.

**Abstract:**

We investigated the role of PI3Kγ in oral carcinogenesis by using a murine model of oral squamous carcinoma generated by exposure to 4-nitroquinoline 1-oxide (4NQO) and the continuous human cancer cell line HSC-2 and Cal-27. PI3Kγ knockout (not expressing PI3Kγ), PI3Kγ kinase-dead (carrying a mutation in the PI3Kγ gene causing loss of kinase activity) and wild-type (WT) C57Bl/6 mice were administered 4NQO via drinking water to induce oral carcinomas. At sacrifice, lesions were histologically examined and stained for prognostic tumoral markers (EGFR, Neu, cKit, Ki67) and inflammatory infiltrate (CD3, CD4, CD8, CD19 and CD68). Prevalence and incidence of preneoplastic and exophytic lesions were significantly and similarly delayed in both transgenic mice versus the control. The expression of prognostic markers, as well as CD19^+^ and CD68^+^ cells, was higher in WT, while T lymphocytes were more abundant in tongues isolated from transgenic mice. HSC-2 and Cal-27 cells were cultured in the presence of the specific PI3Kγ-inhibitor (IPI-549) which significantly impaired cell vitality in a dose-dependent manner, as shown by the MTT test. Here, we highlighted two different mechanisms, namely the modulation of the tumor-infiltrating cells and the direct inhibition of cancer-cell proliferation, which might impair oral cancerogenesis in the absence/inhibition of PI3Kγ.

## 1. Introduction

Phosphoinositide 3-kinases (PI3Ks) are a group of eight plasma membrane-associated lipid kinases grouped into three classes (based on their primary structure, regulation, and in vitro lipid substrate specificity) [1]. Class I kinases received great attention because of their involvement in important processes such as cell proliferation and survival [2]: they are heterodimers composed by a 110-kDa catalytic subunit (p110 α, β, γ, δ) complexed with a regulatory part, which allows the interaction with membrane receptors. The main product of class I PI3Ks is phosphatidylinositol-3,4,5-trisphosphate (PIP3): it initiates one of the most important signaling pathways essential for cell growth, proliferation, survival, and migration downstream of growth factors and oncoproteins. Class I PI3Ks are further subgrouped into class IA and IB. The class IA catalytic subunits (p110α, p110β and p110δ) are bound to a p85 regulatory subunit; the class IB catalytic subunit p110γ binds one of two non-p85 regulatory subunits, called p101 and p84. Distinct expression patterns are shown in the four different class I PI3K isoforms [3].

Among the many processes controlled by PI3Ks [4], one of the most captivating is their involvement in cancer development because of the ability of PI3K to trigger a complex panoply of responses impinging on cell survival and proliferation, as well as on the microenvironment [5,6]. The PI3K signaling pathway is believed to be deregulated in a wide spectrum of human cancers [7], and genetic analysis has shown that the PI3Kα plays a dominant role in the most common human neoplasm, such as breast, colon, gastric, cervical, prostate, and lung cancer [8,9,10]. Isoforms β and δ also seem to be involved in some tumors [11,12,13]. The fourth member of the class I PI3K subgroup, PI3Kγ, is abundantly expressed in immune cells of myeloid origin, which regulate innate immunity in both inflammation and cancer [14,15,16], but its role in tumors is still puzzling. Efimenko and colleagues demonstrated the importance of PI3Kγ in T-cell acute lymphoblastic leukemia progression [17], and an elevated expression of p110γ has been reported in chronic myeloid leukemia [18] as well as in invasive breast carcinoma [19]. The expression of p110γ was upregulated in renal carcinoma cell lines, compared to an immortalized proximal tubule epithelial cell line from a normal adult human kidney [20]. Edling and colleagues reported that p110γ expression is increased in pancreatic ductal adenocarcinoma tissue compared with normal ducts, and that its downregulation through siRNA reduces cell proliferation, highlighting a critical role for p110γ in pancreatic cancer progression [21]. A high-throughput mutational analysis identified novel somatic mutations affecting p110γ in different types of tumors, including breast, lung, ovarian, and prostate cancer [22].

Nevertheless, to the best of our knowledge very few studies have been carried out on PI3Kγ involvement in oral squamous cells carcinoma (OSCC) [23,24]. It is the most common oral malignancy [25] whose therapeutic outcomes are currently still limited, mainly due to its special location, delayed diagnosis and relapses, as well as poor understanding of the underlying molecular mechanism [26]. Oral carcinogenesis is mainly caused by tobacco and alcohol consumption, and numerous inflammation-mediated molecular pathways have been explored and studied as important events in promoting oral carcinogenesis.

With these premises in mind, we decided to investigate the role of PI3Kγ in a murine model of OSCC generated by exposure to the chemical carcinogen 4-nitroquinoline 1-oxide (4NQO) that produces close similarity with human OSCC at both histological and molecular levels [27,28,29,30,31]. The use of 4NQO is widely recognized as a surrogate of tobacco exposure to tissues of the aerodigestive tract. The study has been conducted on PI3Kγ kinase-dead mice (PI3Kγ^KD/KD^, mice carrying a targeted point mutation in the PI3Kγ gene causing loss of lipid kinase activity) and on PI3Kγ knockout mice (PI3Kγ^−/−^, mice with a deletion of the PI3Kγ protein) [32]. Moreover, we analyzed PI3Kγ expression and inhibition in three human cell lines derived from oral cavities, two neoplastic and one represented by continuous keratinocytes.

## 2. Materials and Methods

### 2.1. Materials

All reagents were purchased from Sigma (St. Louis, MO, USA) unless otherwise stated.

### 2.2. Animals

The study was conducted according to the guidelines of the Declaration of Helsinki and approved by the Italian Ministry of the Health, protocol code 625/2017-PR, date of approval 2 August 2017.

Twenty PI3Kγ knockout mice and twenty PI3Kγ kinase-dead mice in a C57Bl/6 background were generated as previously described [33]. Twenty-five age-matched C57Bl/6 mice were used as controls. Experiments were performed on three-month-old male mice. All animals were maintained at standard laboratory conditions of alternating 12 h periods of light and darkness. The ambient temperature was 29 ± 1 °C during the whole experimental period. Neither PI3Kγ^−/−^ nor PI3Kγ^KD/KD^ transgenic mice ever displayed spontaneous development of oral tumors [33,34].

### 2.3. Chemically Induced Carcinogenesis and Lesion Development

Mice were administered 4NQO via drinking water (0.1 g/L) ad libitum to induce oral carcinomas [27,35]. After 9 weeks of 4NQO administration, the oral cavity of each mouse was examined under light anesthesia every second week. The lesions were counted, measured, scored and photographed. The end-points for data analysis included prevalence and multiplicity of preneoplastic (OPLs) and exophytic lesions (ExLs). Total lesions covered all the lesion types, while different kinds of leukoplakia were considered preneoplastic lesions (OPLs). Prevalence indicated the percentage of mice with lesions, and multiplicity represented the average number of lesions carried by each mouse. A “pathological score” (PS), expressing the overall situation of every single animal, was the sum of the score of every single lesion present in the oral cavity, based on the double-blind scoring of lesions as previously described [36] according to the following rules: “0” for no lesions, “1” or “2” for a whitish tongue (depending on the severity), “3” for any OPL, “4” to “6” for every ExL according to the diameter (“4”: ExL with a diameter < 1 mm; “5”: ExL with a diameter between 1 and 3 mm; “6”: ExL with a diameter > 3 mm). Mice were euthanized after 22 weeks of 4NQO-exposure for initial suffering of the control group, accordingly with the OECD (Guidance Document on the Recognition, Assessment, and Use of Clinical Signs as Humane Endpoints for Experimental Animals Used in Safety Evaluation). Animals dead before 22 weeks were not included in the experimentation.

### 2.4. Histological and Immunohistochemical Analysis

After sacrifice, tongues were immediately removed, fixed in 4% paraformaldehyde in phosphate-buffered saline (PBS) for 3 h, washed in PBS and embedded in paraffin after dehydration with ascending ethanol passages (50, 70, 80, 95, 100%) followed by diaphanization in Bioclear (Bio-Optica, Milano, Italy). To identify all the lesions, tongues were sectioned completely (7 µm thick), from end to end, using an RM2135 microtome (Leica Microsystems); sections were placed on slides and put into a drying oven overnight. One slide every fifteen was then deparaffinated and rehydrated with decreasing ethanol passages and stained with hematoxylin and eosin (H&E) (Carlo Erba Reagents, Milan, Italy); the slides were immersed in 0.1% hematoxylin for 10 min, washed in tap water for 15 min, immersed in 0.1% eosin for 5 min, and washed in distilled water. The sections were then dehydrated with ascending ethanol passages and mounted in Dibutylphthalate Polystyrene Xylene (DPX). According to the histological features, lesions were classified into dysplasia (low, mild, high grade), and in situ or invasive carcinoma. Immunohistochemistry staining was performed using IHC Select^®^ HRP/DAB (Merck Millipore, Burlington, MA USA) according to manufacturer instructions. Briefly, after being deparaffinized, slides were treated with 0.1% trypsin solution to recover tissue antigenicity. Then, 3% hydrogen peroxide solution was used to block endogenous peroxidase activity. After an incubation of 5 min in Blocking Reagent, primary antibodies (listed in Table 1) were left to incubate overnight at 4 °C. The next day, the secondary antibody provided by the kit was added to the slices for 10 min, sequentially followed by incubation with streptavidin HRP (10 min) and with the chromogen reagent (8 min). To counterstain tissues, slides were treated with hematoxylin dye for 1 min, dehydrated and covered with a coverslip using DPX.

### 2.5. Cell Culture

HSC-2 (human cell line derived from oral squamous cell carcinoma), Cal-27 (human oral adenosquamous carcinoma cell line) and SG (human gingival epithelioid cell line) were kindly provided by Prof. Harvey Babich (Yeshiva University, New York, NY, USA), while HeLa (human cell line derived from cervical cancer) and 293T (human cells derived from fetal kidney, expressing SV40 large T antigen), representing, respectively, the positive and the negative control for PI3Kγ expression, were generously provided by Prof. Riccardo Autelli (University of Turin, Turin, Italy).

HSC-2, Cal-27 and SG cells were cultured in RPMI-1640 medium (PAA Laboratories GmbH, Cölbe, Germany), while HeLa and 293T were grown in DMEM, both supplemented with 10% fetal calf serum (FCS, PAA Laboratories GmbH), 100 U/mL penicillin G, 40 μg/mL gentamicin sulfate and 2.5 μg/mL amphotericin B at 37 °C in a humidified 5% CO_2_ atmosphere.

### 2.6. Immunoblotting

Cells were collected from the culture dish with ice-cold PBS and homogenized in RIPA lysis buffer (150 mM NaCl, 1.0% IGEPAL^®^ CA-630, 0.5% sodium deoxycholate, 0.1% SDS, 50 mM Tris, Sigma-Aldrich, Merck KgaA, Darmstadt, Germany) supplemented with a protease inhibitor cocktail (Cell Signalling, Thermo Fisher Scientific, Rodano, Milan, Italy). Samples were treated as previously described [37]. Thirty μg of total protein extracts were then separated by 7.5% SDS-PAGE. After transfer, the membrane was incubated overnight with primary antibody, mouse anti-PI3Kγ (Santa Cruz Biotechnology sc-166365, Dallas, TX, USA), at 4 °C. The membrane was then washed three times and incubated with an anti-mouse secondary antibody conjugated with HRP (1:5000, Immunological Sciences, Rome, Italy) for 1 h at room temperature. The blot was further washed three times and images were visualized with the ChemiDoc™ Touch Imaging System Bio-Rad.

### 2.7. Cell Viability Assay

Cell viability assay was performed as previously described [36]. Briefly, cells were grown on 96-well plates at a density of 1 × 10^4^ cells/cm^2^. After 24 h, the cells were exposed to increasing concentrations of specific PI3Kγ inhibitor IPI-549 (DBA Italia, Milan, Italy), or vehicle (DMSO) as control. Cell viability was measured by MTT assay after 24 h of treatment. Experiments were repeated three times in octuplicate.

### 2.8. Statistics

Cell viability and histological results were analyzed by one-way ANOVA followed by Tukey’s multiple comparison post hoc test. Lesion multiplicity and PS among different groups at different times were compared with two-way ANOVA followed by the Bonferroni post hoc test. Fisher’s exact test was used for lesion prevalence comparisons. Statistical analysis was performed by the IBM SPSS program 24.0 version. A difference with *p* < 0.05 was considered significant.

## 3. Results

### 3.1. Chemically Induced OSC Carcinogenesis

The OSC carcinogenesis followed the multistep process as previously described by Tang et al. [35]; at the end of the experimental period, all 4NQO-exposed control animals had developed lesions. Six control animals and two in each group among the transgenic mice died during the induction period.

### 3.2. Oral 4NQO-Carcinogenesis Is Delayed in PI3Kγ^KD/KD^ and PI3Kγ^−/−^ Mice

OPL prevalence and multiplicity (Figure 1A,B) in PI3Kγ^KD/KD^ and PI3Kγ^−/−^ mice were comparable and both significantly lower than control mice before the 19th week of treatment. The absence of PI3Kγ (PI3Kγ^−/−^ mice) or of its lipid kinase activity (PI3Kγ^KD/KD^) delayed the development of OPLs during the exposure to 4NQO: between week 11 and 15 (Figure 1A), the difference between control and transgenic mice was of utmost significance, since at least 70% of WT mice showed OPLS, while both PI3Kγ^KD/KD^ and PI3Kγ^−/−^ mice had null or scarce lesions. From week 15, both transgenic strains started developing OPLs, while preneoplastic lesion number in control mice decreased, probably due to their transformation into EXLs (Figure 1C). An analogous trend was observed for OPL multiplicity, the most significant difference between control and transgenic mice being among the 11th and the 19th weeks (Figure 1B).

Consistently with the chemical multistep carcinogenetic model, ExLs followed the preneoplastic lesions appearing around the 15th week of exposure to 4NQO. From the 17th week onwards, a sharp increase in ExL prevalence (Figure 1C) and multiplicity (Figure 1D) was observed in control mice. At week 19, about 30% of WT mice showed ExLs, while only 10% of PI3Kγ^KD/KD^ and no PI3Kγ^−/−^ had ExLs; prevalence in controls reached 100% in the following two weeks. Only 40% PI3Kγ^KD/KD^ and 20% PI3Kγ^−/−^ mice displayed ExLs at sacrifice (Figure 1C). Moreover, on average, more than twice as many ExLs were found in WT compared to PI3Kγ^KD/KD^ and PI3Kγ^−/−^ animals at the end of the experimental period (Figure 1D). Differences between transgenic and WT mice were even more striking when considering both total lesion prevalence and multiplicity (Figure 1E,F), which were significantly delayed in PI3Kγ^KD/KD^ and PI3Kγ^−/−^.

The average lesion-free time was longer in PI3Kγ^KD/KD^ and PI3Kγ^−/−^ mice. In comparison with WT mice, the development of total lesions and OPLs was delayed for 7–9 weeks (*p* < 0.01), while ExL appearance was delayed for at least 2 weeks in PI3Kγ^KD/KD^ and PI3Kγ^−/−^ mice.

When considering the overall situation of each oral cavity, pathological scoring confirmed that carcinogenesis was delayed when PI3Kγ is absent or inactive, whereas no significant difference was detected between PI3Kγ^KD/KD^ and PI3Kγ^−/−^ mice (Figure 2).

PI3Kγ^KD/KD^ and PI3Kγ^−/−^ mice showed similar responses to 4NQO exposure: no significant difference in lesion development was detected between these two groups.

### 3.3. Lesion Severity Is Decreased in PI3Kγ^KD/KD^ and PI3Kγ^−/−^ Mice if Compared with WT Mice

To better understand the overall carcinogenesis and the severity of the lesions in each mouse tongue, sections from control, PI3Kγ^KD/KD^ and PI3Kγ^−/−^ 4NQO-exposed mice were analyzed by H&E staining.

At sacrifice, PI3Kγ^KD/KD^ and PI3Kγ^−/−^ tongues showed mid–low degree dysplasia and only one case of invasive carcinoma was observed in either PI3Kγ^KD/KD^ or PI3Kγ^−/−^ mice. On the other hand, 13 out of 25 WT mice displayed mid–high degree dysplasia and OSCC (Figure 3A). Figure 3B shows representative histological features unveiled at sacrifice of a SCC in a WT tongue and two different low–mid dysplasia developed in PI3Kγ^KD/KD^ and PI3Kγ^−/−^ mice (Figure 3C,D).

### 3.4. Immunohistochemical Analysis of Prognostic-Related Biomarker of Oral Cancerogenesis

Table 2 depicts the results of IHC staining of WT, PI3Kγ ^KD/KD^ and PI3Kγ^−/−^ lesions. EGFR expression is decreased in mutated mice, both in preneoplastic and OSCC. Neu and Ki67 expression is also higher in control mice masses, despite being more evident in OSCC and in preneoplastic lesions, respectively. cKit is faintly more expressed in wild type (WT).

### 3.5. Immunohistochemical Characterization of Infiltrating Immune Cell Subsets in Tongues

CD3^+^ (pan-T), CD4^+^ (T-helper) and CD8^+^ (T cytotoxic) cells were more abundant in tongues isolated from PI3Kγ^KD/KD^ and PI3Kγ^−/−^ mice than in wild type. CD19^+^ (pan-B) and namely CD68^+^ (pan-macrophages) cells were, on the other hand, more numerous in wild-type tongues (Table 3).

### 3.6. PI3Kγ Expression in Neoplastic and Epithelioid Oral Cell Lines and the Effect of Its Inhibition on Cell Vitality

We explored PI3Kγ expression in OSCC (HSC-2 and Cal-27) and epithelioid (SG) cell lines by Western blot analysis (Figure 4A). HeLa and 293T cells were included, respectively, as positive and negative controls. A band of 110 KDa was detected in all tested oral cell lines, though PI3Kγ expression in SG was fainter.

The PI3Kγ-specific inhibitor IPI-549 strikingly impaired HSC-2 and Cal-27 cells vitality (Figure 4B) already after 24 h treatment: their CC_50_ being 32.61 and 55.75 µM, respectively. Interestingly, the vitality of the oral non-neoplastic cell line SG and that of cells not expressing PI3Kγ (293T) was significantly less affected (CC_50_ > 160 µM, undetected in the concentration range tested).

## 4. Discussion

Our preclinical study demonstrates that PI3Kγ absence reduces the number and delays the appearance of chemically induced oral preneoplastic and neoplastic lesions, therefore supporting its role in oral cancer development. Some other authors already showed that PI3K isoforms other than γ are involved in oral cancerogenesis [38,39,40,41], and our results therefore confirm the PI3K/Akt/mTOR pathway as a potential target.

The identification of novel molecular targets has been fundamental for the great improvements in oncology, as shown by the identification of Herb2 and the use of drugs against it in the treatment of Herb2^+^ breast cancer. Despite some interesting progress achieved in oncology over the past 25 years, OSCC remains one of the most challenging cancers to be cured and the pharmacological approach is often ineffective [42], namely because of the lack of specific molecular targets.

With these premises, we wondered if PI3Kγ might be looked at as a target in OSCC treatment. Here, we analyzed the role of PI3Kγ on chemically induced oral carcinogenesis in a genetically modified background [43]. This approach represents a valid method to study the function of specific genes in vivo.

Morphological and histological changes induced in control mice (C57Bl/6 wild type) by 4NQO also appeared in the PI3Kγ^KD/KD^ and PI3Kγ^−/−^. Thus, we could compare the differences among groups and investigate the role of PI3Kγ on preneoplastic lesions and OSCC initiation and promotion. Reduced lesion prevalence and multiplicity, decreased PS, and lower incidence of OSCC versus controls were equally found in PI3Kγ^KD/KD^ and PI3Kγ^−/−^ mice. Since PI3Kγ is involved in two distinct signaling pathways (a kinase-dependent activity that controls phosphorylation of its substrate, and a kinase-independent activity that relies on protein interactions) [32], our results suggest that kinase activity, but not the scaffold function of PI3Kγ, is fundamental in facilitating tumor progression. This is supported as well by a lower expression of the tested tumor markers (EGFR, Neu, cKIT and Ki67), both in preneoplastic and frank lesions developed in both PI3Kγ^KD/KD^ and PI3Kγ^−/−^ mice. The expression of those markers in oral carcinogenesis is thoroughly recognized as fundamental both for diagnostic and prognostic purposes. Ki67, whose expression is faint in lesions of our transgenic mice, has been accounted to provide a diagnostic and poor prognostic biomarker for OSCC patients [44]. In a study performed on 102 squamous cell carcinomas of the tongue, the intracytoplasmic expression of Neu, as well as Ki-67 nuclear staining, have been associated with a high risk of recurrence of tongue OSCC [45]. EGFR is more expressed both in preneoplastic and neoplastic lesions from transgenic mice, consistently with its recognized role in oral carcinogenesis; moreover, recently it has also been considered as a possible predictor of metastasis [46].

Since PI3Kγ expression was firstly detected in leukocytes [47], great attention has been devoted to its role in creating a microenvironment favoring angiogenesis, tumor growth, and immunosuppression. Immune cell infiltration is an important feature of oral cancer, and tumor progression reflects the inability of the immune system to recognize and eliminate neoplastic cells [48]. Therefore, we characterized the inflammatory infiltrated in WT and transgenic mice: the higher presence of CD3^+^ (pan T-cell), CD4^+^ (T-helper cell), and overall CD8^+^ (cytotoxic T lymphocytes) in PI3Kγ^KD/KD^ and PI3Kγ^−/−^ mice, where cancer lesions were less precocious and less numerous than in control mice, corresponds to what is reported in the literature. A high number of CD8^+^ infiltrating cells correlates with favorable outcomes in patients since they migrate into the peritumoral region and directly fight tumor cells [48] and the number of peritumoral CD8^+^ T cells correlated with a lower neoplastic expression of Ki67 in the work published by Öhman et al. [49], consistently with what we found in our model. Moreover, low levels of CD4^+^ and CD8^+^ lymphocytes have been found in patients with oral cancer with active disease, thus suggesting a decreased function of effector cells. An elevated CD3 expression in infiltrating lymphocytes was as well considered an independent factor for favorable prognosis [50]. On the other hand, the pan-B cell marker CD19 and the pan macrophage marker CD68 were less expressed in PI3Kγ^KD/KD^ and PI3Kγ^−/−^ mice than in control mice. The role of CD19^+^ cells in tumor immunosurveillance is quite debated, as reviewed by Hadler-Olsen and Wirsing [51], since only in some works their presence has been significantly correlated to survival benefits. On the other hand, coherently with what we found, CD68^+^ cells seem to be associated with poor clinical outcomes [52], though no definite position on the role of CD68^+^ cells in oral cancer immune surveillance seems to be achieved [51].

Strict evidence demonstrates the presence of immunosurveillance already on dysplasia/preneoplastic lesions in which, as for frank lesions, T cells play a major role impairing the neoplastic transformation [49,53]. To this phenomenon, as well as to the well-known ability of PI3Kγ in inducing immune suppression through AKT, mTOR and NfKβ pathways [15], we might ascribe the high number of OPL taking over the very few OSCC that we observed in transgenic mice.

PI3kγ expression has been recently shown to be not only restricted to leukocytes, but also to other cell types, including tumor cells [17,18,19,20], suggesting that it might be involved in roles other than driving the tumor microenvironment. This intriguing hypothesis has been considered in our work: we demonstrated that cells from human OSCC (HSC-2 and Cal-27) express PI3Kγ and that its inhibition through the specific antagonist IPI-549 impairs cell proliferation significantly more than that of 293T PI3Kγ-negative cells. Despite being preliminary and deserving further studies, the effect of the PI3Kγ inhibitor on the vitality of human continuous keratinocytes from the oral cavity (SG) is less striking than what was observed in cancer cells, thus suggesting possible lower cytotoxicity on oral keratinocytes.

## 5. Conclusions

Based on our results and on those of other authors demonstrating that PI3Kγ plays a proinflammatory role [54], we can suggest that the antitumoral activity observed when PI3Kγ is absent (or inhibited) is not only due to the modulation of the tumor “inflamed” microenvironment, but also to a direct impact on cancer cell survival. The next fundamental step will be the comprehension of the mechanisms by which PI3Kγ inhibition impairs cancer cell proliferation.

If the promising results obtained in our model are confirmed in humans, PI3Kγ inhibition might be proposed to prevent progression to OSCC in patients with OPLs and those already operated for OSCC with a consequently high risk of recurrence or relapse. PI3Kγ, differently from p110α or p110β [55], is not required for normal development, lifespan, or basic immune responses, unless under stress conditions; therefore, low toxicity and few side effects are expected by acting on PI3Kγ as a therapeutic target [56].

## Figures and Tables

**Figure 1 cancers-13-04211-f001:**
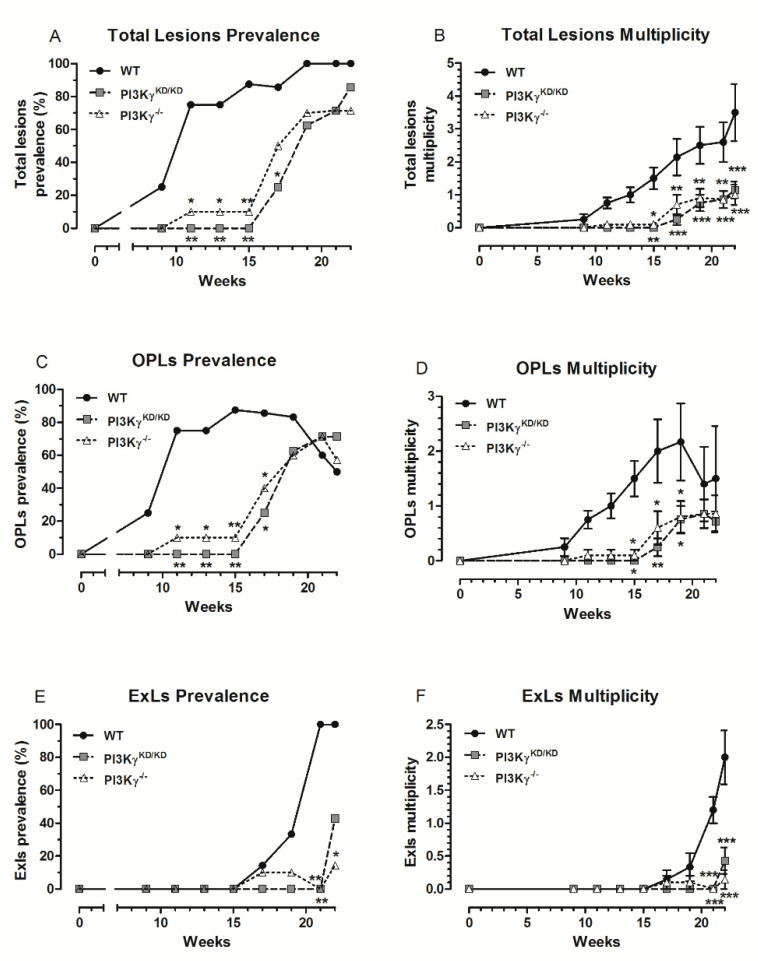
(**A**) Total lesion prevalence, (**B**) total lesion multiplicity, (**C**) OPL prevalence, (**D**) OPL multiplicity, (**E**) ExL prevalence, and (**F**) ExL multiplicity of WT, PI3Kγ^KD/KD^ and PI3Kγ^−/−^ mice treated with 4NQO. Values represent mean ± SEM. PI3Kγ^KD/KD^/PI3Kγ^−/−^ vs. WT: * *p* < 0.05, ** *p* < 0.01 and *** *p* < 0.001.

**Figure 2 cancers-13-04211-f002:**
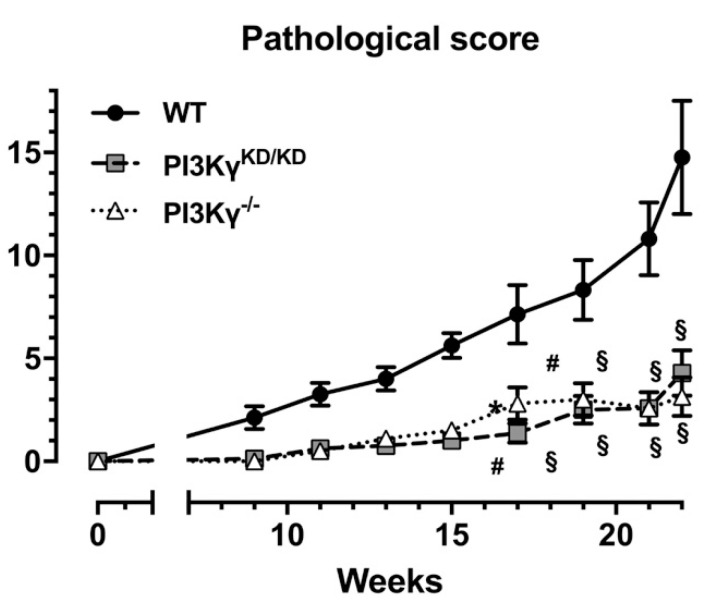
Mean pathological score (as the sum of the score of every single lesion per animal) of WT, PI3Kγ^KD/KD^ and PI3Kγ^−/−^ mice receiving 4NQO treatment are shown. PI3Kγ^KD/KD^/PI3Kγ^−/−^ vs. WT: * *p* < 0.05, # *p* < 0.01 and § *p* < 0.001.

**Figure 3 cancers-13-04211-f003:**
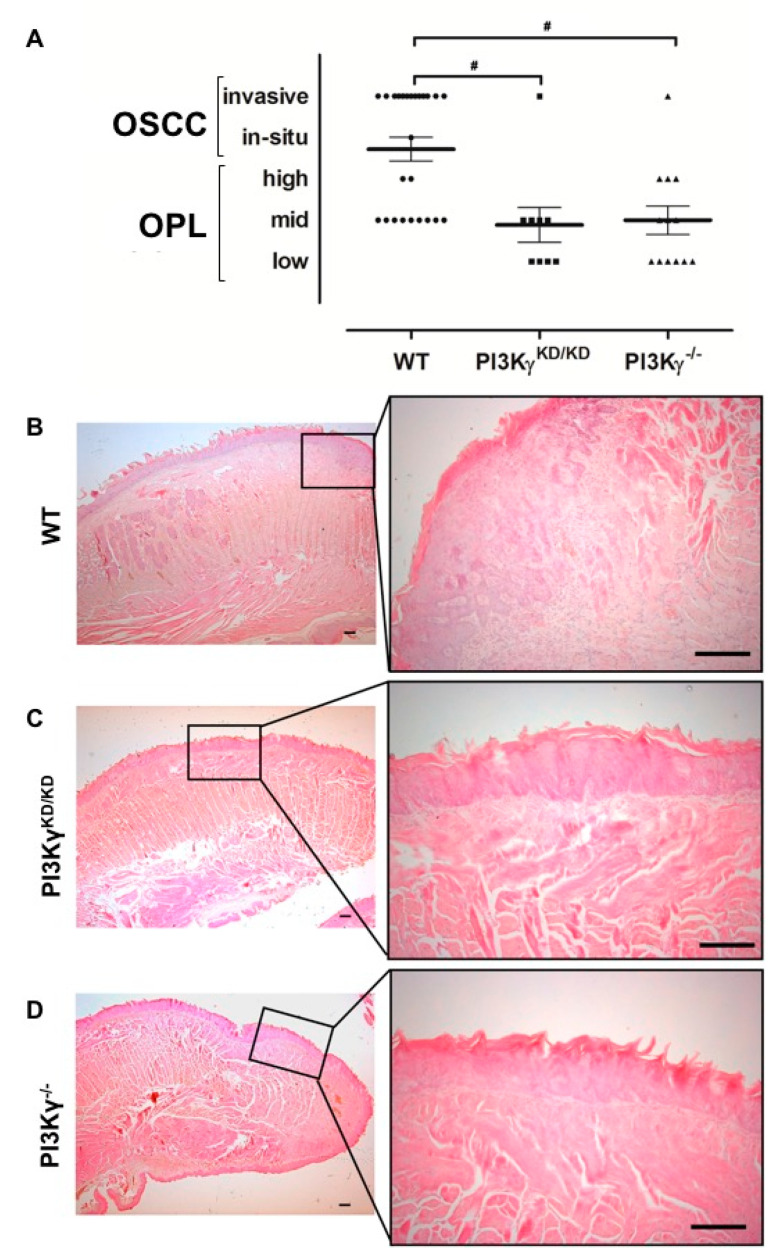
(**A**) Histological analysis: global overview of WT, PI3Kγ^KD/KD^ and PI3Kγ^−/−^ mice lesions at sacrifice. (**B**–**D**) Representative H&E staining of tongue sections in (**B**) WT, (**C**) PI3Kγ^KD/KD^ and (**D**) PI3Kγ^−/−^ mice at the end of 4NQO treatment (22 weeks). Values represent mean ± SEM. PI3Kγ^KD/KD^/PI3Kγ^−/−^ vs. WT: # *p* < 0.01. Scale bars, 100 μm.

**Figure 4 cancers-13-04211-f004:**
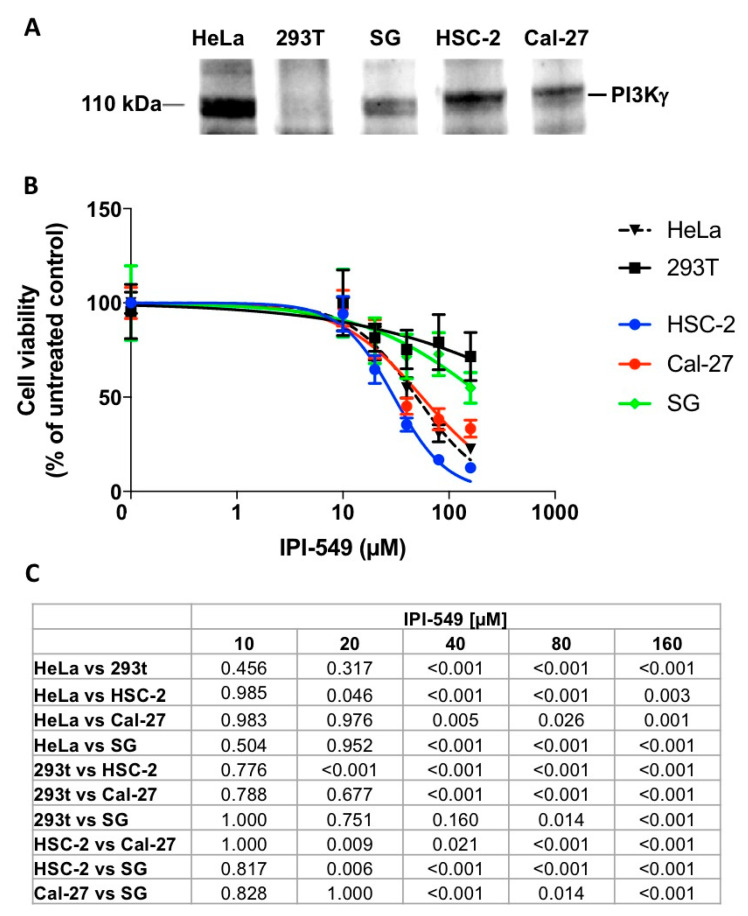
(**A**) Western blot analysis of PI3Kγ expression in HeLa, 293T, SG, HSC-2 and Cal-27 whole cell lysates. HeLa and 293T represent, respectively, the positive and the negative control for PI3Kγ expression. (**B**) Residual cell vitality in presence of the PI3Kγ specific inhibitor IPI-549 (0–160 µM) as detected by MTT assay. Values, expressed as a percentage of controls, represent mean ± standard error (SEM), *p* are detailed in (**C**).

**Table 1 cancers-13-04211-t001:** Antibodies used in immunohistochemistry.

Primary Ab (Clone)	Host	Dilution	Supplier
CD3 (PC3/188A)	Mouse	1:200	Santa Cruz Biotechnology
CD4	Rabbit	1:200	Abbiotec
CD8	Rabbit	1:200	Abbiotec
CD19	Rabbit	1:300	Abbiotec
CD68	Rabbit	1:200	Abbiotec
EGFR clone 8G6.2	Mouse	1:100	Merck Millipore
c-ErbB2/c-Neu (Ab-5)	Mouse	1:100	Calbiochem
Ki67 (H-300)	Rabbit	1:200	Santa Cruz Biotechnology
c-Kit (C-19)	Rabbit	1:100	Santa Cruz Biotechnology

**Table 2 cancers-13-04211-t002:** Prognostic-related biomarker expression in 4NQO-induced preneoplastic lesions and OSCC isolated from wild type (WT), PI3Kγ ^KD/KD^ (KD) and PI3Kγ^−/−^ (KO) mice.

Marker	Preneoplastic Lesions		OSCC	
	WT	KD and KO	WT	KD and KO
EGFR	++ ^1^	+/–	+++	+
NEU	+	+/–	++	+
cKit	+	+/–	++	+
Ki67	+++	+	++	+

^1^ +/−, barely present; +, faintly present; ++, present; +++, abundant.

**Table 3 cancers-13-04211-t003:** Abundance of infiltrating immune cell subsets in tongues isolated from wild-type, PI3Kγ^KD/KD^ (KD) and PI3Kγ^−/−^ (KO) mice.

Infiltrating Cells	Wild Type	KD and KO
CD3	+/– ^1^	+
CD4	+/–	+
CD8	–	+
CD19	++	+
CD68	++	+/–

^1^ –, Absent; +/–, barely present; +, faintly present; ++, present.

## Data Availability

https://figshare.com/articles/dataset/Chemical_oral_cancerogenesis_is_impaired_in_PI3K_knockout_and_kinase-dead_mice/15028740.

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
