# Peer review of "Chemical Oral Cancerogenesis Is Impaired in PI3Kγ Knockout and Kinase-Dead Mice"

_cancers, 2021, doi:10.3390/cancers13164211_

Round 1
Reviewer 1 Report
In the research article titled “Chemical oral cancerogenesis is impaired in PI3Kγ knockout 2 and kinase-dead mice”. The authors investigated the role of PI3Kγ in oral carcinogenesis by using a murine model of oral squamous carcinoma generated by exposure to 4-nitroquinoline 1-oxide (4NQO) and the continuous human cancer cell line HSC-2. PI3Kγ knockout (not expressing PI3Kγ), PI3Kγ kinase-dead (carrying a mutation in the PI3Kγ gene causing loss of kinase activity), and wild type (WT) C57Bl/6 mice were administered 4NQO via drinking water to induce oral carcinomas. They demonstrated that the absence/inhibition of PI3Kγ impairs oral carcinogenesis through two different mechanisms acting synergistically, namely the modulation of the tumor-infiltrating cells and the direct inhibition of cancer proliferation. Although the study is very small but very logical. Overall, the study is interesting and I am sure there will be more experiments needed. The authors need to thoroughly check the article for grammatical and typo errors to make it coherent and comprehensive.
Therefore, for these reasons and other issues listed below, I suggest reconsidering this article for publication in cancers after revision.
Issues and Comments:
- I think the author should correct the sentence in line 44
- The author should correct the cell number with a superscript in line 190.
- I am sure the author will provide better picture quality in the final version of the manuscript.
- I think all the experiments were performed logically, however the data are small, I leave this concern to the editorial office to decide whether the manuscript with 4 figures fits the standard of the journal.
Author Response
Authors’ responses to Reviewers:
Dear Editor,
Thank you for the comments regarding our manuscript (cancers-1324822) entitled “Chemical oral cancerogenesis is impaired in PI3Kγ knockout and kinase-dead mice” by G.N Berta et al.
The Reviewers comments spurred us to further improve the overall quality of our message.
Yours faithfully,
Giovanni N Berta
Reviewer #1
Comment 1 to the Authors: I think the author should correct the sentence in line 44
Authors’ response/action: We are grateful to the Reviewer for rising this point: the typo has been amended.
Comment 2 to the Authors: The author should correct the cell number with a superscript in line 190.
Authors’ response/action: We apologize for the typo; it has been amended (now line 217).
Comment 3 to the Authors: I am sure the author will provide better picture quality in the final version of the manuscript.
Authors’ response/action: We optimized the resolution of the pictures
Reviewer 2 Report
Several issues arise that make the manuscript not qualified for publication at the present form. 1. How to determine “multiplicity”? The authors should clearly describe. 2. The authors should provide the data to confirm that the loss of kinase activity of their “kinase-dead” transgenic mice. For example, performing immunoprecipation of PI3Kr protein and do the in vitro phosphorylation assay with a suitable substrate. 3. To claim the protective role of immune cell infiltration to the oral carcinogenesis in PI3K knockout mice, only examining the immune cell types within tumor lesions is not sufficient. The authors should use depletion strategy to provide the direct evidence of which immune cell type plays the most important role of protection. 4. In the last part of the examining of PI3Kr inhibitor to the proliferation of OSCC cells, the purpose of HeLa cells, a cervical cancer cell line, used was not clear. Furthermore, the authors should add at least one more OSCC cell line to their analysis. The authors should use normal oral keratinocytes as the control rather than 293T cells. 5. The authors should consider to use IPI-549 in 4-NOQ induced OSCC model, which will have the clinical sound than knockout or KD transgenic mice models.Author Response
Dear Editor,
Thank you for the comments regarding our manuscript (cancers-1324822) entitled “Chemical oral cancerogenesis is impaired in PI3Kγ knockout and kinase-dead mice” by G.N Berta et al.
The Reviewers comments spurred us to further improve the overall quality of our message.
Yours faithfully,
Giovanni N Berta
Reviewer #2:
Comment 1 to the Authors: How to determine “multiplicity”? The authors should clearly describe.
Authors’ response/action: as specified in line 135 of the M&M section, “…multiplicity represented the average number of lesions carried by each mouse.”
Comment 2 to the Authors: The authors should provide the data to confirm that the loss of kinase activity of their “kinase-dead” transgenic mice. For example, performing immunoprecipation of PI3Kγ protein and do the in vitro phosphorylation assay with a suitable substrate.
Authors’ response/action: Mice carrying mutations in the Pik3cg locus, like the kinase dead and the knockout strains, were published several years ago. For full characterization of the knockout allele please see Hirsch et al., Science 2000 (PMID 10669418, ref 29 in our manuscript). For full characterization of the kinase-dead mutant you can refer to Patrucco et al., Cell 2004 (PMID 15294162, ref 28 in our manuscript). As indicated, extensive research has been done through the years to prove that the kinase-dead allele lost the catalytic activity but retained distinct scaffolding function. For more recent literature please refer to the review by Ghigo et al., Circulation Research 2017 (PMID 28729453). Some of these papers are now cited in the revision and we believe that further characterization of the mutant strains used in this study is not needed.
Comment 3 to the Authors: To claim the protective role of immune cell infiltration to the oral carcinogenesis in PI3K knockout mice, only examining the immune cell types within tumor lesions is not sufficient. The authors should use depletion strategy to provide the direct evidence of which immune cell type plays the most important role of protection.
Authors’ response/action: we thank the referee for this interesting observation, following which we modified our claim about the putative role of infiltrating cells modulation in our model (lines 44-46). Nevertheless, the demonstration of the protective role of immune cells in PI3K knockout mice carcinogenesis goes beyond the aim of the present manuscript. Several important articles (ref 48-53 in our manuscript) with an approach identical to ours, demonstrated the role of CD3, CD4 and CD8 positive cells in cancer progression, and we referred to those works to show that the composition of the infiltrating population is changed in knockout mice versus wild type. We proposed the observed change in the infiltrating populations as one of the possible mechanisms concurring to the delay in chemical carcinogenesis in our model, also considering the role of PI3Kγ on inflammatory processes.
Comment 4 to the Authors: In the last part of the examining of PI3Kr inhibitor to the proliferation of OSCC cells, the purpose of HeLa cells, a cervical cancer cell line, used was not clear. Furthermore, the authors should add at least one more OSCC cell line to their analysis. The authors should use normal oral keratinocytes as the control rather than 293T cells.
Authors’ response/action: as suggested by the antibody manufacturer, and reported in literature, HeLa cells were used simply as positive controls for western blot analysis investigating PI3Kγ expression, as reported in M&M section (lines 186-189). Anyway, to help the reader, we repeated it in figure 4 legend (lines 351-352).
According to the reviewer suggestion, we performed our experiments also on a second human cell line derived from an oral adenosquamous carcinoma (Cal-27).
Although 293T cells were used as a negative control, as previously reported in literature, we accepted the request of reviewer #2 and added an oral continuous cell line (SG keratinocytes) in our experiments.
New results are shown in figure 4, described in the results section (see point 3.7 of results) and discussed (see last paragraph of discussion).
Comment 5 to the Authors: The authors should consider to use IPI-549 in 4-NOQ induced OSCC model, which will have the clinical sound than knockout or KD transgenic mice models.
Authors’ response/action: One of the major aims of this work was to evaluate the role of the kinase vs. the scaffold function of the PI3Kγ. We would have not been able to discriminate this aspect, which is fundamental for the design of novel inhibitors; therefore, the use of the transgenic mice was necessary. Certainly, the next step will be to evaluate a possible clinical application of our findings, therefore we will certainly consider the reviewer’s suggestions for the continuation of our research.
Reviewer 3 Report
Berta and colleagues performed a basic science study evaluating the effect of PI3K pathway on oral carcinogenesis in a murine model. Specifically, the authors compared the development of preneoplastic and neoplastic squamous cell carcinoma lesions of the oral cavity in PI3K knock out, PI3K kinase-dead and wild type (WT) mice exposed to carcinogens. The authors main findings are that PI3K knock and kinase-dead mice were less likely to develop neoplastic and preneoplastic lesion compared to WT. These findings, although similar in other cancer sites, provide compelling evidence of the role of PI3K in delaying oral carcinogenesis and may provide insight to design human models, potentially patients with oral cavity cancers.
Author Response
Reviewer #3:
No Comments or suggestion for the Authors
Reviewer 4 Report
In this study, the authors evaluated the chemical oral carcinogenesis in PI3Kr knockout and kinase-dead mice. Generally this is potential interesting manuscript. Some revisions are suggested to improve the manuscript.
- For pathological scoring, whether all lesions in the oral cavity (tongue and other sites) were scored?
- For 3-OPLs, whether the size of lesions (e.g., leukoplakia etc.) were analyzed?
- Some crucial papers regarding chemical carcinogenesis of the oral cavity should be introduced (line 85-98) or discussed in the revised manuscript (e.g. In Vivo 2019;33(6):1751-1755; Oral Oncol 2006 May;42(5):448-60; Oral Oncol. 2001 Sep;37(6):477-92; Oral Oncol. 2006 Aug;42(7):655-67 and more others).
- It is better if the authors may evaluate the sequential changes of these proteins in carcinogenesis of animals, but not at end point (22 weeks of exposure to 4-NQO).
- For immunoblotting: Whether the cells (HSC-2, 293T, HeLa) were treated with 4-NQO? The authors may also evaluate the changes of other PI3K isoforms (alpha, better etc.).
- Whether cells can be cultured from different kind of mice (PI3Kr knockout and kinase-dead mice) for in vitro study?
- Figure 2 showed the pathological score (as longitudinal axis unit). The score is higher than 6-7, so the pathological score is the average sum of scores of different lesions in each animal?
- For immunohistochemical staining results in Table 2: any results of control animals (without treatment by 4-NQO)?
- For discussion the role of PI3K in oral cancer, it is better to discuss possible the contribution of various PI3Ks in oral carcinogenesis (e.g., J Oral Pathol Med. 2016 Aug;45(7):469-74.; J Oral Pathol Med. 2016 Nov;45(10):746-752; Aging (Albany NY). 2019 Dec 12;11(23):11624-11639; J Cell Mol Med. 2020 Apr;24(7):4011-4022.).
- Errors in English spelling and grammar can be checked and corrected.
Author Response
Dear Editor,
Thank you for the comments regarding our manuscript (cancers-1324822) entitled “Chemical oral cancerogenesis is impaired in PI3Kγ knockout and kinase-dead mice” by G.N Berta et al.
The Reviewers comments spurred us to further improve the overall quality of our message.
Yours faithfully,
Giovanni N Berta
Reviewer #4:
Comment 1 to the Authors: For pathological scoring, whether all lesions in the oral cavity (tongue and other sites) were scored?
Authors’ response/action: yes, all the lesions in the oral cavity were considered for the entire analysis. Anyway, according to our experience, in this model the lesions appear mostly on the tongue.
Comment 2 to the Authors: For 3-OPLs, whether the size of lesions (e.g., leukoplakia etc.) were analyzed?
Authors’ response/action: we thank the reviewer for his/her precious suggestion because, while reading the M&M section we realized that this aspect was unclear. Any OPL was scored 3, independent from its dimensions. See the paragraph “Chemically-induced carcinogenesis … of Mat and met section (line 137-141).
Comment 3 to the Authors: Some crucial papers regarding chemical carcinogenesis of the oral cavity should be introduced (line 85-98) or discussed in the revised manuscript (e.g. In Vivo 2019;33(6):1751-1755; Oral Oncol 2006 May;42(5):448-60; Oral Oncol. 2001 Sep;37(6):477-92; Oral Oncol. 2006 Aug;42(7):655-67 and more others).
Authors’ response/action: references have been added as suggested (line 102).
Comment 4 to the Authors: It is better if the authors may evaluate the sequential changes of these proteins in carcinogenesis of animals, but not at end point (22 weeks of exposure to 4-NQO).
Authors’ response/action: we are not sure we understood the question. We infer the reviewer refers to the prognostic tumoral markers whose expression has been analyzed at sacrifice. If this is the case, we agree with the reviewer that it would be very interesting to monitor their expression all along the carcinogenetic processes but this goes beyond the aim of this work and this analysis would imply a completely different setting of the study.
Comment 5 to the Authors: For immunoblotting: Whether the cells (HSC-2, 293T, HeLa) were treated with 4-NQO? The authors may also evaluate the changes of other PI3K isoforms (alpha, better etc.).
Authors’ response/action: the aim of the western blot was not to check if 4-NQO modifies the expression of PI3Kg, but to check if cells from OSCC (HSC-2) express PI3Kg in order to justify the results of the proliferation test performed in presence of the PI3Kγ inhibitor. HeLa cells were used simply as positive controls and 293T as negative controls for western blot analysis investigating PI3Kγ expression, as suggested by antibody manufacturers and detailed in the M&M section. Anyway, to help the reader, we repeated it in figure 4 legend (lines 351-352). The study of changes of other PI3K isoforms goes, though absolutely interesting, goes beyond the aim of our work, as explained in the introduction.
Comment 6 to the Authors: Whether cells can be cultured from different kind of mice (PI3Kγ knockout and kinase-dead mice) for in vitro study?
Authors’ response/action: we agree with the reviewer on the importance of using cells from transgenic mice to crosscheck the in vivo experiments. We have repeatedly tried to isolate and cultivate cells from transgenic mice, but, unfortunately, since cells were not immortalized, they did not live long enough to perform all the experiments. Therefore, we had to fall back on continuous human OSCC cells.
Comment 7 to the Authors: Figure 2 showed the pathological score (as longitudinal axis unit). The score is higher than 6-7, so the pathological score is the average sum of scores of different lesions in each animal?
Authors’ response/action: we apologize for the lack of clarity; we have better explained the scoring in the M&M section (lines 137-138) and also amended the legend of figure 2 (line 281).
Comment 8 to the Authors: For immunohistochemical staining results in Table 2: any results of control animals (without treatment by 4-NQO)?
Authors’ response/action: the request is more than licit, and when we started using this model we performed this kind of control on untreated animals. Though, no inflammatory infiltrate in untreated mucosa could be observed, therefore we considered, for our present work, 4-NQO treated wild type mice as the most appropriate control.
Comment 9 to the Authors: For discussion the role of PI3K in oral cancer, it is better to discuss possible the contribution of various PI3Ks in oral carcinogenesis (e.g., J Oral Pathol Med. 2016 Aug;45(7):469-74.; J Oral Pathol Med. 2016 Nov;45(10):746-752; Aging (Albany NY). 2019 Dec 12;11(23):11624-11639; J Cell Mol Med. 2020 Apr;24(7):4011-4022.).
Authors’ response/action: thank you for the suggestion, the role of PI3K isoforms in oral cancer has been discussed, and the references proposed by the reviewer have been introduced (see line 359-361).
Comment 10 to the Authors: Errors in English spelling and grammar can be checked and corrected.
Authors’ response/action: errors have been amended
Round 2
Reviewer 2 Report
The authors have done several improvements to their manuscript and it could be accepted in this present form.
Reviewer 4 Report
improve much.